# Gender-Dependent Associations between Serum Betatrophin Levels and Lipoprotein Subfractions in Diabetic and Nondiabetic Obese Patients

**DOI:** 10.3390/ijms242216504

**Published:** 2023-11-19

**Authors:** Hajnalka Lőrincz, Sára Csiha, Balázs Ratku, Sándor Somodi, Ferenc Sztanek, Ildikó Seres, György Paragh, Mariann Harangi

**Affiliations:** 1Division of Metabolism, Department of Internal Medicine, Faculty of Medicine, University of Debrecen, 4032 Debrecen, Hungary; 2Doctoral School of Health Sciences, University of Debrecen, 4032 Debrecen, Hungary; 3Department of Emergency Medicine, Faculty of Medicine, University of Debrecen, 4032 Debrecen, Hungary; 4Institute of Health Studies, Faculty of Health Sciences, University of Debrecen, 4032 Debrecen, Hungary

**Keywords:** betatrophin, angiopoietin-like protein 8, triglyceride, lipoprotein subfractions, high-density lipoprotein, diabetes, obesity

## Abstract

Betatrophin, also known as angiopoietin-like protein 8 (ANGPTL8), mainly plays a role in lipid metabolism. To date, associations between betatrophin and lipoprotein subfractions are poorly investigated. For this study, 50 obese patients with type 2 diabetes (T2D) and 70 nondiabetic obese (NDO) subjects matched in gender, age, and body mass index (BMI) as well as 49 gender- and age-matched healthy, normal-weight controls were enrolled. Serum betatrophin levels were measured with ELISA, and lipoprotein subfractions were analyzed using Lipoprint gel electrophoresis. Betatrophin concentrations were found to be significantly higher in the T2D and NDO groups compared to the controls in all subjects and in females, but not in males. We found significant positive correlations between triglyceride, very low density lipoprotein (VLDL), large LDL (low density lipoprotein), small LDL, high density lipoprotein (HDL) -6-10 subfractions, and betatrophin, while negative correlations were detected between betatrophin and IDL, mean LDL size, and HDL-1-5. Proportion of small HDL was the best predictor of betatrophin in all subjects. Small LDL and large HDL subfractions were found to be the best predictors in females, while in males, VLDL was found to be the best predictor of betatrophin. Our results underline the significance of serum betatrophin measurement in the cardiovascular risk assessment of obese patients with and without T2D, but gender differences might be taken into consideration.

## 1. Introduction

Betatrophin, also known as TD26, lipoprotein lipase (LPL) inhibitor (lipasin), and angiopoietin-like protein 8 (ANGPTL8), is a protein that is primarily expressed in the liver and the adipose tissues [1]. Based on former studies, it mainly plays a role in lipid metabolism [2]. Betatrophin shares homology with other members of the ANGPTL family, especially with angiopoietin-like protein 3 (ANGPLT3)’s N-terminal domain, which regulates blood lipids, and with angiopoietin-like protein 4 (ANGPLT4)’s N-terminal segment, which mediates LPL binding [1]. Previous studies demonstrated increased concentrations of betatrophin in metabolic diseases associated with lipid disorders, such as dyslipidemia [3] and non-alcoholic fatty liver disease [4].

Obesity is characterized by excessive accumulation of body fat leading, to many medical complications, including type 2 diabetes mellitus (T2D) or impaired glucose tolerance and dyslipidemia [5]. However, overweight and obese individuals demonstrate significant heterogeneity of obesity phenotypes, which might be related to the participation of organokines, genes, and cells in addition to environmental, social, and economic factors [6]. Moreover, gender may also affect clinical appearance. Nondiabetic obese (NDO) individuals can be characterized by metabolic disturbances, including atherogenic dyslipidemia and early changes in organokine expression, but their glucose metabolism is still within normal ranges. Therefore, NDO represents an intermediate status between healthy subjects and obese T2D patients with altered organokine patterns, chronic inflammation, and increased cardiovascular risk [5,7]. Despite a lack of clear definitions for classifying obesity subgroups [8], there are some promising markers that are useful in making basic differentiations between these phenotypes [5].

Insulin resistance/hyperinsulinemia is the most common metabolic disorder in obesity, and it is the main driving force behind the development of dyslipidemia. High concentrations of triglycerides accompanied by decreased high-density lipoprotein cholesterol (HDL-C) concentrations are its main characteristics. Low-density lipoprotein cholesterol (LDL-C) concentrations could be optimal or mildly increased, although the number of small dense LDL particles can be increased [9]. However, components of atherogenic dyslipidemia are detected in NDO patients as well [10]. Indeed, previous studies reported elevated serum levels of betatrophin in both T2D [11,12,13,14] and in obese patients without T2D [14,15], but data on the association between lipid parameters and betatrophin are incongruent and show gender differences. The associations between betatrophin and lipoprotein subfractions are poorly investigated. In a recent prospective case–control study involving 41 pregnant women with and without gestational diabetes mellitus (GDM), serum levels of LDL6 subfraction, betatrophin, and the potent LPL activator glycosylphosphatidylinositol-anchored high-density lipoprotein-binding protein 1 were found to be higher in the GDM group compared to the controls, which may be part of an adaptive mechanism in response to insulin resistance. However, significant correlations between LDL subfractions and betatrophin were not detected [16].

Therefore, we aimed to determine the circulating betatrophin levels in NDO patients, obese T2D patients, and healthy control subjects. The potential correlations between anthropometric parameters, markers of carbohydrate metabolism, LDL and HDL subfractions, and serum betatrophin levels were also investigated in the three cohorts. We hypothesized that betatrophin correlates with atherogenic lipoproteins, including triglyceride-rich lipoproteins, and may also show gender-specific differences.

## 2. Results

Anthropometric data, main medications, and the routine laboratory parameters are presented in Table 1. All three groups are matched in age and gender. Waist circumference and body mass index (BMI) were normal in healthy controls, while they were comparably high in T2D and NDO groups. Significantly lower HDL-C and ApolipoproteinA-I (ApoA1) levels were observed in T2D and NDO subjects compared to the controls, while total cholesterol and LDL-C levels did not differ significantly. Furthermore, the concentrations of triglyceride, high-sensitivity C-reactive protein (hsCRP), fasting glucose, hemoglobin A1C (HbA_1C_), insulin, and supersensitive thyroid stimulating hormone (sTSH) were found to be significantly higher in the T2D and NDO groups compared to the controls. It must be noted that fasting glucose, insulin and HbA_1C_ levels were in normal range in NDO and control patients, while TSH levels were normal in all patient groups.

The proportion and amount of very-low-density lipoprotein (VLDL) subfraction were higher in T2D subjects compared to NDO and controls. The proportion of large LDL subfraction was significantly higher in both T2D and NDO compared to controls, while higher amount of large LDL subfractions were found in NDO than in controls. The proportion and amount of intermediate density lipoprotein (IDL) and small LDL were comparable in the three cohorts. Furthermore, the mean LDL size was significantly lower in the T2D and NDO groups compared to the controls (Table 2). We detected a shift from large-sized to small-sized HDL subfractions in both obese groups. The proportion and amount of large HDL, as well as the amount of intermediate HDL were significantly lower in T2D and NDO subjects. Whereas the proportion and amount of small HDL were also higher in the T2D and NDO groups compared to the controls (Table 2).

Median betatrophin concentrations were found significantly higher in T2D and NDO groups compared to controls (T2D: 26.85 (19.00–37.10) vs. NDO: 18.22 (13.14–26.12) vs. controls: 15.35 (11.72–23.88) ng/mL; *p* = 0.00004 between T2D vs. controls; and *p* = 0.0061 between NDO vs. controls, respectively) (Figure 1a). Similar tendencies were detected in females (T2D: 27.89 (20.71–44.19) vs. NDO: 18.15 (13.61–28.71) vs. controls: 14.36 (9.88–21.50) ng/mL; *p* = 0.00001 between T2D vs. controls; and *p* = 0.0056 between NDO vs. controls, respectively) (Figure 1b) but not in males (T2D: 21.58 (15.62–31.84) vs. NDO: 21.18 (11.16–25.36) vs. controls: 20.08 (14.48–24.73) ng/mL; *p* = 1.0 between T2D vs. controls; and *p* = 0.7 between NDO vs. controls, respectively) (Figure 1c). Analyzing all subjects, we did not find significant difference regarding serum betatrophin levels in females and in males (18.76 (13.30–29.20) vs. 20.76 (12.80–26.40) ng/mL; *p* = 0.87) (Figure 1d).

Betatrophin showed significant positive correlations with BMI, waist circumference, HbA_1C_, insulin, hsCRP, and triglyceride in overall subjects (Figure 2a) and in females (Figure 2b). In males, only insulin levels correlated with betatrophin of these variables (Figure 2c). There were positive correlations between proportion of VLDL, large LDL, small LDL and betatrophin, while the proportion of IDL correlated negatively with betatrophin in overall subjects (Figure 2a) and in females (Figure 2b). In males, only the proportion of VLDL and the proportion of IDL showed correlation with betatrophin (Figure 2c). In addition, mean LDL size correlated negatively with betatrophin in all participants and females, but not in males (Figure 3a–c).

Analyzing data of HDL subfraction tests, HDL-1 to -5 subfractions showed significant negative correlations with betatrophin in overall subjects and in females (Figure 2a,b). Whereas strong positive correlations were found between betatrophin and HDL-6 to -10 in all subjects and in females (Figure 2a,b). Similar but weaker correlations were observed between HDL-2, -3, -7, -8, -9 and betatrophin in males (Figure 2c).

There was a negative correlation between large HDL and betatrophin in all subjects (Figure 3d), in females (Figure 3e), and in males (Figure 3f). In turn, a strong positive correlation was detected between small HDL and betatrophin in all subjects (Figure 3g) and in females (Figure 3h), but not in males (Figure 3i).

We analyzed these correlations not only in overall subjects, but also in the T2D, NDO and in the control group and summarized the results in the Appendix A. We observed various correlations between betatrophin and laboratory parameters in subgroups. In patients with T2D, serum betatrophin showed significant correlations with BMI, waist circumference, log10 insulin, log10 triglyceride, IDL, and partially with different HDL subfractions (HDL-2, -3, -7, and -8). However, in NDO group, betatrophin did not correlate with BMI, log10 insulin, or log10 triglyceride and only partially correlated with lipoprotein subfractions (Appendix A).

Furthermore, we analyzed the correlations of conventional cardiovascular risk factors (age, BMI, CRP, insulin and HOMA-IR) and lipoprotein subfractions. The results are summarized in Appendix A. HsCRP showed significant correlations with triglyceride-rich lipoprotein subfractions and small HDL, while BMI, insulin, and HOMA-IR correlated mainly with HDL subfractions.

Backward stepwise multiple regression analyses were performed to determine significant predictor(s) of betatrophin in overall subjects and in different subgroups (results are presented in Table 3). The variables of BMI, HbA_1C_, triglyceride; VLDL (%); IDL (%); large LDL (%); small LDL (%), large HDL (%); and small HDL (%) were included in all models. Proportion of small HDL was the best predictor of betatrophin in all subjects (standardized β = 0.446; *p* < 0.0001). The best predictors of betatrophin were small LDL (standardized β = 0.378; *p* < 0.0001) and large HDL subfractions (standardized β = −0.330; *p* < 0.0001) in females; while it was the VLDL (standardized β = 0.425; *p* = 0.006) in males that indicated important gender differences in the regulatory role of betatrophin. Interestingly, the best predictors of circulating betatrophin levels were different in the three patient cohorts. According to our data, betatrophin level was best predicted by the large HDL in T2D, small LDL in NDO, and small HDL in the controls (Table 3).

## 3. Discussion

Betatrophin (ANGPLT8) is the most recently identified member of the ANGPLT proteins, expressed primarily in the liver [17]. Betatrophin acts as a hepatokine in an autocrine/paracrine manner, decreasing intracellular triglyceride content of hepatocytes through the downregulation of lipogenic transcription factors [4]. Former studies reported that higher serum betatrophin levels are associated with insulin resistance in T2D patients [11,12,13,18]. Moreover, elevated serum levels of betatrophin were associated with higher risk of all-cause mortality in T2D [19]. In line with these data, we found significantly higher serum levels of betatrophin in the T2D cohort compared to the NDO patient group and the control population in the whole population and in females, but not in males. Former studies found increased circulating levels of betatrophin in human obesity [14]. In another previous study, betatrophin levels were found to be markedly elevated in nondiabetic obese female subjects, but not in males [15]. In our NDO patients, serum betatrophin was slightly elevated compared to the control group in the whole study population and in females, but these differences were not significant.

Recent studies have proved that betatrophin has a key regulatory role in triglyceride metabolism through its interaction with ANGPLT3. The ANGPLT8/ANGPLT3 complex is required to intracellular secretion and co-folding. Indeed, this complex has an increased ability to bind LPL compared to ANGPLT3 or betatrophin alone [20]. Furthermore, the complex formation can unmask the inhibitory domain of LDL. Interestingly, betatrophin is also expressed in the white adipose tissue, where ANGPLT3 is not expressed [1]. Here the betatrophin interacts with ANGPLT4 forming a complex, which has a poorer LPL inhibitory effect compared to ANGPLT4 alone, indicating the ANGPLT4-neutralizing effect of betatrophin [21]. A recent review has summarized the latest findings and provided new evidence for an updated ANGPTL8/ANGPTL3/ANGPTL4 models related to triglyceride partitioning between adipose and oxidative tissues [22].

Serum betatrophin positively correlated with triglyceride in nondiabetic subjects and negatively correlated with HDL-C levels in nondiabetic subjects and T2D patients [23,24]. Moreover, significant positive correlation was found between serum betatrophin and triglyceride levels in a cohort including control subjects and patients with insulin resistance and sepsis [25] and in patients with metabolic syndrome [26]. We also found significant positive correlations between triglyceride and betatrophin levels, but only in the entire patient population and in females. It must be noted that HDL metabolism is strongly affected by hypertriglyceridemia associated with diabetes and obesity. The increased number of triglyceride-rich lipoproteins results in increased cholesteryl ester transfer protein activity, which exchanges cholesterolesters from HDL for triglyceride from VLDL and LDL, leading to the formation of smaller, cholesteryl-ester-depleted, and triglyceride-enriched HDL particles [27]. Furthermore, lipolysis of triglyceride-rich HDL occurs by hepatic lipase resulting in small HDL with a reduced affinity for ApoA-I, which leads to dissociation of ApoA-I from HDL. This will ultimately lead to lower levels of HDL-C and a reduction in circulating HDL particles with impairment of reversed cholesterol transport [28]. Consequently, correlation between triglyceride and betatrophin levels explains the association between HDL size and betatrophin.

Recent studies have suggested that subfraction measurements of lipoproteins may improve the estimation of cardiovascular risk and may lead to further improvements in identification of appropriate targets for therapeutic intervention in individual patients [29]. The number of small LDL particles has been reported to be higher in patients with coronary artery disease as compared to healthy subjects. Similarly, concentration of large HDL subfraction was noted to have a strong inverse correlation with the risk of acute coronary syndrome [30]. Accordingly, increased proportions of small LDL and HDL subfractions represent an atherogenic lipoprotein profile [31,32].

Although the data of some in vitro and in vivo animal studies have been published, this is the first clinical study evaluating the serum levels of betatrophin and its correlations with lipoprotein subfractions in patients with T2D as well as NDO and control subjects. We detected strong negative correlation between the proportion of large HDL and betatrophin in the whole study cohort, in females and males, and between mean LDL size and betatrophin in the whole study population and in females. Significant positive correlation was found between the proportion of small HDL and betatrophin in all patients and in females. Positive correlations were found between VLDL and betatrophin in the whole study population and in both genders. These findings are in line with the data of Chen et al. [21], who reported that the ANGPLT8/ANGPLT3 complex blocked LPL-facilitated hepatocyte VLDL-cholesterol uptake in vitro. Therefore, elevated levels of VLDL may be found in circulation, which correlates with betatrophin levels. Based on our data, betatrophin correlates with the atherogenic lipoprotein subpopulations; therefore, measurement of circulating betatrophin may be used in cardiovascular risk assessment in obese and non-obese subjects, especially in females.

The gender differences in betatrophin levels and correlations between lipid parameters, lipoprotein subfractions, and betatrophin must be emphasized. Although atherosclerotic cardiovascular disease is the leading cause of death among women, compared to men, they are less likely to be diagnosed appropriately and to be provided with adequate treatment and preventive care [33]. Therefore, identifying gender-specific risk factors may be particularly useful when developing strategies for the prevention of cardiovascular diseases. Estrogens are now known to have potent anti-atherogenic properties through lipid and non-lipid mechanisms. Although the exact mechanism is not fully elucidated, higher levels of estrogen might be responsible for the different lipoprotein metabolism including the higher LPL activity in females [34]. Furthermore, estrogen increases HDL-C and decreases both LDL-C and lipoprotein (a). Gender difference in betatrophin levels might be the possible link between female gender and altered lipoprotein metabolism. These data might also have therapeutic implications. On one hand, the benefit of estrogen replacement therapy in cardiovascular prevention [35] can be revised and personalized based on concomitant diseases and betatrophin levels. On the other hand, the efficacy of novel angiopoietin-like protein inhibitors [36] might be evaluated separately in female patients with dyslipidemia.

Some limitations of the study must be mentioned. Although the studied populations are relatively large—we enrolled a total of 169 subjects—enrollment of even larger populations could improve the power of the study. Furthermore, the lower ratio of males in all study cohorts may limit the interpretation of gender differences. Although our observational study cannot clarify the exact role of betatrophin in lipid metabolism, it may encourage further research to explore the possible link between the circulating betatrophin levels and the development of an atherogenic lipid profile. The clinical relevance of gender difference in betatrophin levels also requires further investigation.

Our results underline the importance of studying the effect of betatrophin on the metabolism of atherogenic lipoprotein subfractions in a complex way in severe obesity with and without T2DM, especially in females.

## 4. Materials and Methods

### 4.1. Enrolment of Study Participants

For this study, 50 obese patients with T2D (20 males and 30 females; mean age: 47.6 ± 8.1 years; mean BMI: 43.14 ± 9.34 kg/m^2^) and 70 NDO patients (19 males and 51 females; mean age: 44.8 ± 12.0 years; mean BMI: 43.3 ± 7.9 kg/m^2^) matched in gender, age, and BMI were enrolled from the diabetes and obesity outpatient clinics at the Department of Internal Medicine, Faculty of Medicine, University of Debrecen, Hungary. We also enrolled 49 healthy gender- and age- matched normal-weight subjects (16 males and 33 females; mean age: 43.2 ± 9.1 years; mean BMI: 24.7 ± 2.8 kg/m^2^) as controls to this observational study. Obesity was defined as BMI ≥ 30 kg/m^2^. All participants provided written informed consent. Permission to carry out this study was granted by the Regional Ethics Committee of the University of Debrecen and the Medical Research Council (registration numbers DE RKEB/IKEB 5513B-2020 and IV/7989-1/2020/EKU, respectively).

Exclusion criteria were age below 18 years and active liver, kidney, gastrointestinal, pulmonary, neurological, autoimmune, acute infective, or endocrine diseases (except for T2D). Further exclusion criteria were pregnancy, lactation, smoking, and regular alcohol consumption. The subjects were referred to scheduled medical appointments in the morning and arrived after an overnight fast. Peripheral venous blood samples were collected into Vacutainer^®^ serum separator tubes and EDTA-anticoagulated tubes, respectively (Becton Dickinson, San Jose, CA, USA), and centrifuged after 30 min coagulation at 3500× *g* 10 min +4 °C.

### 4.2. Determination of Routine Laboratory Parameters

The routine laboratory parameters, including total cholesterol, triglyceride, HDL-C, LDL-C, ApoAI, triglyceride, hsCRP, fasting glucose, HbA_1C_, insulin, glomerular filtration rate, aspartate aminotransferase, and sTSH, were measured immediately from the same vendor at the Department of Laboratory Medicine, University of Debrecen using Roche reagents (Roche, Basel, Switzerland) and appliances from the same manufacturer (Roche Cobas 600 analyzer, Basel, Switzerland). Furthermore, 0.5 mL aliquots of serum and plasma samples were kept frozen at −80 °C for enzyme-linked immunosorbent assay (ELISA) measurements and for lipoprotein subfraction analysis.

### 4.3. Determination of Serum Betatrophin

Concentrations of betatrophin were measured via 5-fold dilution as duplicate using a commercially available ELISA (human betatrophin ELISA, Cat: RD191347200R, BioVendor, Brno, Czech Republic) according to the user manual. The intra- and inter-assay variation coefficients were 5.4–9.4% and 2.4–5.3%, respectively. The limit of detection was 0.244 ng/mL.

### 4.4. Determination of Lipoprotein Subfractions

Lipoprotein subfractions were determined using the Lipoprint^®^ System (Quantimetrix Corporation, Redondo Beach, CA, USA) using LDL and HDL subfraction tests according to the instructions of the manufacturer as described previously [7]. The lipoprotein subfractions were distributed based on their size. Briefly, 25 µL of the sample was pipetted into polyacrylamide gel tubes, and Sudan Black containing loading gel as lipophilic dye was used. The prefilled tubes were photopolymerized for 30 min and then electrophorized with 3 mA/tube for 1 h. Each electrophoresis chamber involved a quality control provided by the manufacturer (Liposure Serum Lipoprotein Control, Quantimetrix Corp., Redondo Beach, Cerritos, CA, USA). Tubes were scanned within 2.5 h with an ArtixScan M1 digital scanner (Microtek International Inc., CA, USA). The subfraction bands were analyzed with the Lipoware Image SXM v.1.82 Software (Quantimetrix Corp., Redondo Beach, CA, USA).

During LDL subfraction analyses, three IDL bands (Midbands C through A) and up to seven LDL subfraction can be distributed among VLDL (Rf = 0) and HDL (Rf = 1) bands in the densitograms. Percentage of large LDL (large LDL%) was defined as the summed percentages of LDL1 and LDL2, whereas percentage of small LDL (small LDL%) was defined as the sum of LDL3–LDL7. Cholesterol concentrations of LDL subfractions were determined by multiplying the relative area under the curve (AUC) of subfractions by total cholesterol concentration. Mean LDL size was also calculated with Lipoware Image SXM v.1.82 Software. The intra-assay precisions were 0.58–7.28% for VLDL, 3.85–11.14% for Midbands, and 1.05–1.52% for LDL. The inter-assay precisions were 7.12–9.40% for VLDL, 7.47–10.90% for Midbands, and 1.26–1.57% for LDL.

During HDL subfraction analyses, 10 HDL subfractions can be detected between VLDL + LDL (Rf = 0) and albumin (Rf = 1) bands. HDL1–3 corresponded to large HDL, HDL4–7 corresponded to intermediate HDL, and HDL8–10 corresponded to small HDL subfractions. Cholesterol contents of HDL subfractions were calculated using AUC by multiplying the HDL-C of the samples. The intra- and inter-assay precisions were 0.90–1.47% and 2.49–4.75%, respectively.

### 4.5. Statistical Analyses

Statistical analyses were performed using Statistica 13.5.0.17 software (TIBCO Software Inc., Tulsa, OK, USA). Figures were prepared using GraphPad Prism 6.01 software (GraphPad Prism Software Inc., San Diego, CA, USA). Normality of continuous data was accessed with a Kolmogorov–Smirnov test. Data were presented as mean ± SD or median (interquartile range, IQR). The difference between gender ratios was calculated with the Chi-squared test. Comparison of study groups was performed with one-way ANOVA (Tukey’s post hoc test) in case of normally distributed variables and with a Kruskal–Wallis H test in case of variables with non-normal distributions. Pearson’s correlation was used to investigate the relationship between selected variables. Logarithmic transformation (log10) was carried out before correlation analyses of variables with non-normal distribution. To determine significant predictor(s) of betatrophin, backward stepwise multiple regression analysis was performed. Results were considered to be significant at *p* < 0.05.

## 5. Conclusions

We demonstrated strong correlations between atherogenic lipoprotein subfractions and betatrophin levels in T2D and NDO patients as well as in control subjects. The increase in serum betatrophin levels in T2D and NDO patients was gender-dependent and predominantly observed in female subjects. The pattern of correlations between lipoprotein subfractions and betatrophin levels were also gender-dependent, indicating the possible effect of estrogens on the regulatory role of betatrophin in lipoprotein metabolism. These results may improve our knowledge on the characteristics of lipoprotein metabolism in females. Our results may underline the significance of serum betatrophin measurement in cardiovascular risk assessment of obese patients with and without T2D, but gender differences might be taken into consideration.

## Figures and Tables

**Figure 1 ijms-24-16504-f001:**
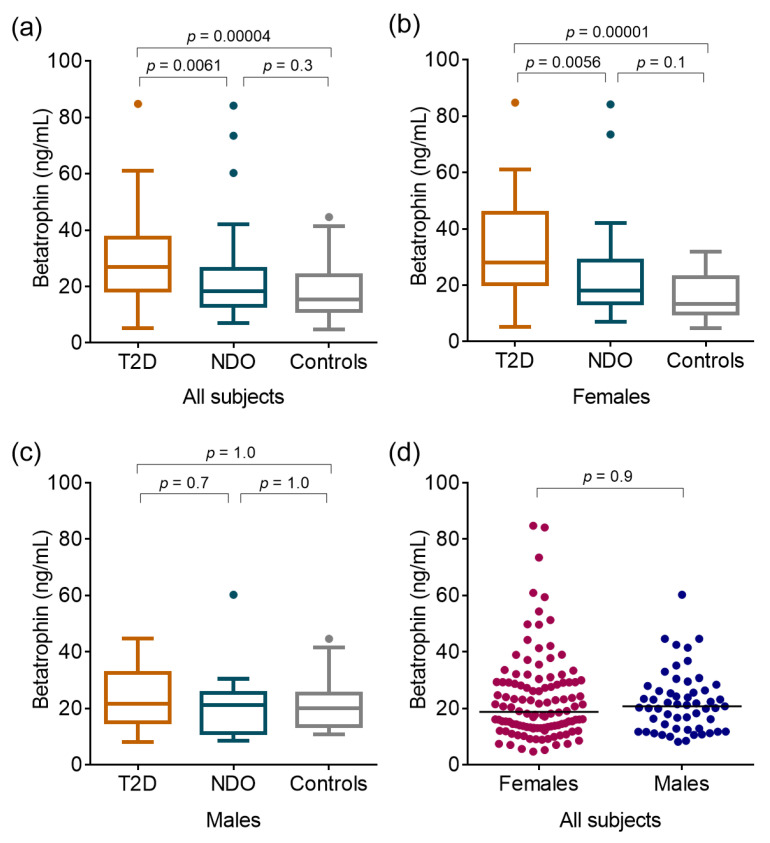
Serum concentrations of betatrophin (**a**) in all enrolled subjects, (**b**) in females, and (**c**) in males. Serum concentrations of betatrophin in females and males in all cohorts (**d**). Notes: Betatrophin levels were determined from peripheral blood samples via enzyme-linked immunoassay. (**a**–**c**): Differences between obese patients with type 2 diabetes (T2D), non-diabetic obese (NDO), and age- and gender-matched normal-weight controls were calculated using a Kruskal–Wallis H test. Boxes represent 25th–75th percentiles, the 50th percentile (median) is shown as a solid line within the boxes, whiskers represent the minimum and maximum levels, excluding outliers indicated with dots. d: Solid lines represent 50th percentile (median).

**Figure 2 ijms-24-16504-f002:**
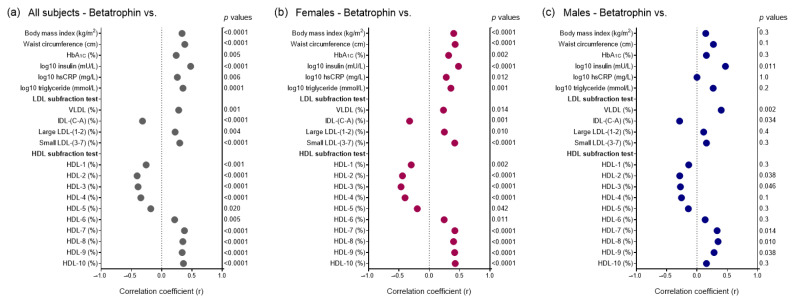
Data of Pearson’s correlations between main anthropometric and laboratory parameters, lipoprotein subfractions, and betatrophin (**a**) in all enrolled subjects, (**b**) in females, and (**c**) in males. Betatrophin levels were determined from peripheral blood samples via enzyme-linked immunoassay, and lipoprotein subfractions were measured using Lipoprint^®^ gel electrophoresis. Abbreviations HbA_1C_, hemoglobin A1C; HDL, high-density lipoprotein; hsCRP, high sensitivity C-reactive protein; IDL, intermediate density lipoprotein; LDL, low-density lipoprotein; VLDL, very-low density lipoprotein.

**Figure 3 ijms-24-16504-f003:**
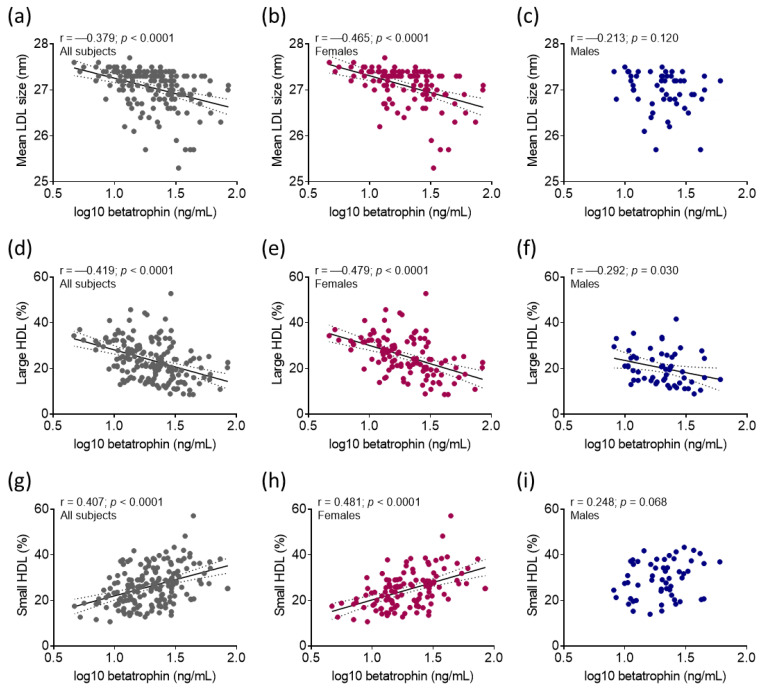
Pearson’s correlation between mean low-density lipoprotein (LDL) size and betatrophin (**a**) in all subjects, (**b**) in females, and (**c**) in males. Pearson’s correlation between the proportion of large high-density lipoprotein (HDL) subfraction and betatrophin (**d**) in all subjects, (**e**) in females, and (**f**) in males. Pearson’s correlation between the proportion of small HDL subfraction and betatrophin (**g**) in all subjects, (**h**) in females, and (**i**) in males. Notes: Pearson’s correlation was used to investigate the relationship between selected variables. Logarithmic transformation (log10) was carried out before correlation analyses of variables with non-normal distribution. Best fit line is represented with black solid line and 95 % confidence interval is represented with black dotted lines.

**Table 1 ijms-24-16504-t001:** Anthropometric data, main medication, and laboratory parameters of enrolled subjects.

	T2D (*n* = 50)	NDO (*n* = 70)	Controls (*n* = 49)
Anthropometric parameters			
Male/Female (*n*)	20/30	19/51	16/33
Age (yrs)	47.6 ± 8.1	44.8 ± 12.0	43.2 ± 9.1
Waist circumference (cm)	127.1 ± 18.5 *	125.5 ± 17.9 §	85.2 ± 12.3
Body mass index (kg/m^2^)	43.1 ± 9.3 *	43.3 ± 7.9 §	24.7 ± 2.8
Main medication			
Metformin (*n*; %)	37; 74.0	8; 11.4	0; 0.0
Insulin (*n*; %)	12; 24.0	0; 0.0	0; 0.0
GLP-1 RA (*n*; %)	9; 18.0	0; 0.0	0; 0.0
Statin (*n*; %)	23; 46.0	8; 11.4	0; 0.0
ACEI/ARB (*n*; %)	24; 48.0	23; 32.9	1; 2.0
CCB (*n*; %)	13; 26.0	6; 8.6	1; 2.0
Diuretics (*n*; %)	8; 16.0	18; 25.7	0; 0.0
Laboratory parameters			
Total cholesterol (mmol/L)	5.0 ± 1.2	5.0 ± 0.9	5.0 ± 0.8
HDL-C (mmol/L)	1.2 ± 0.3 *	1.3 ± 0.3 §	1.5 ± 0.4
LDL-C (mmol/L)	3.0 ± 0.9	3.2 ± 0.8	3.0 ± 0.5
ApolipoproteinA-I (g/L)	1.4 ± 0.2 *	1.5 ± 0.2	1.6 ± 0.3
Triglyceride (mmol/L)	1.7 (1.2–2.7) *	1.4 (1.0–1.9)	1.1 (0.9–1.5)
hsCRP (mg/L)	6.6 (3.1–13.7) *	7.8 (3.2–13.9) §	1.3 (0.6–2.5)
Fasting glucose (mmol/L)	6.3 (5.5–10.5) *#	5.2 (4.9–5.8) §	4.8 (4.5–5.1)
HbA_1C_ (%)	7.2 ± 1.6 *#	5.6 ± 0.9	5.1 ± 0.3
Insulin (mU/L)	25.4 (14.1–31.4) (*n* = 16) *	15 (11.7–21.8) (*n* = 59) §	10.9 (6.6–12.9) (*n* = 16)
GFR (mL/1.73 m^2^)	90 (90–90)	90 (90–90)	90 (90–90)
AST (U/L)	25.0 (17.0–30.5)	20.0 (17.0–27.0)	19.0 (17.0–24.0)
sTSH (mU/L)	2.3 (1.2–15.2) (*n* = 31) *	2.1 (1.6–2.8)	1.7 (1.2–2.1)

Abbreviations: ACEI/ARB, angiotensin-converting enzyme inhibitors/angiotensin II receptor blockers; AST, aspartate aminotransferase; CCB, calcium channels blockers; GLP-1 RA, glucagon-like peptide-1 receptor agonists; HbA_1C_, hemoglobin A1C; HDL-C, high-density lipoprotein cholesterol; hsCRP, high-sensitivity C-reactive protein; GFR, glomerular filtration rate; LDL-C, low-density lipoprotein cholesterol; NDO, nondiabetic obese patients; sTSH, supersensitive thyroid stimulating hormone; T2D, patients with type 2 diabetes. Notes: Data are presented as mean ± SD or median (IQR). Difference of genders was analyzed using Chi-squared test. Comparisons between groups were performed with one-way ANOVA (Tukey’s post hoc test) in the case of normally distributed variables and with a Kruskal–Wallis H test in the case of variables with non-normal distribution. (* indicates *p* < 0.05 between T2D vs. controls; § indicates *p* < 0.05 between NDO vs. controls; # indicates *p* < 0.05 between T2D vs. NDO).

**Table 2 ijms-24-16504-t002:** Proportion and amounts of lipoprotein subfractions in enrolled subjects.

	T2D (*n* = 50)	NDO (*n* = 70)	Controls (*n* = 49)
LDL subfraction test			
VLDL (%)	21.03 ± 5.47 *#	18.30 ± 3.81	17.69 ± 3.21
(mmol/L)	1.07 ± 0.51 *#	0.91 ± 0.23	0.89 ± 0.19
IDL (%)	24.62 ± 3.66	26.14 ± 4.00	26.56 ± 6.34
(mmol/L)	1.23 ± 0.33	1.28 ± 0.30	1.34 ± 0.41
Large LDL (%)	26.58 ± 5.68 *	28.68 ± 5.01 §	23.18 ± 6.10
(mmol/L)	1.32 ± 0.42	1.45 ± 0.38 §	1.17 ± 0.38
Small LDL (%)	1.50 (0.0–3.30)	0.00 (0.00–1.80)	0.60 (0.00–1.90)
(mmol/L)	0.067 (0.000–0.202)	0.028 (0.000–0.103)	0.032 (0.093–0.173)
Mean LDL size (nm)	26.85 (26.60–27.30) *	27.20 (26.90–27.40) §	27.30 (27.00–27.40)
HDL subfraction test			
Large HDL (%)	19.06 ± 6.55 *#	23.47 ± 7.49 §	28.98 ± 8.67
(mmol/L)	0.24 ± 0.12 *	0.30 ± 0.12 §	0.46 ± 0.26
Intermediate HDL (%)	49.02 ± 4.02	50.49 ± 3.96	50.35 ± 4.59
(mmol/L)	0.58 ± 0.15 *#	0.66 ± 0.17 §	0.73 ± 0.17
Small HDL (%)	31.91 ± 7.97 *#	26.04 ± 7.15 §	20.67 ± 6.29
(mmol/L)	0.37 ± 0.10 *	0.34 ± 0.12 §	0.29 ± 0.07

Abbreviations: HDL, high-density lipoprotein; IDL, intermediate-density lipoprotein; LDL, low-density lipoprotein; NDO, nondiabetic obese patients; T2D, patients with type 2 diabetes; VLDL, very low-density lipoprotein. Notes: Serum lipoprotein subfractions were detected using Lipoprint^®^ acrylamide gel electrophoresis. Data are presented as mean ± SD or median (IQR). Comparisons between groups were analyzed with one-way ANOVA (Tukey’s post hoc test) in the case of normally distributed variables and with Kruskal–Wallis H test in the case of variables with non-normal distribution. (* indicates *p* < 0.05 between T2D vs. controls; § indicates *p* < 0.05 between NDO vs. controls; # indicates *p* < 0.05 between T2D vs. NDO).

**Table 3 ijms-24-16504-t003:** Determination of predictor(s) of betatrophin as a dependent variable using backward stepwise multiple regression analysis in different subgroups.

	Predictor	β	*p*
Model 1			
All subjects	Small HDL (%)	0.446	<0.0001
Model 2			
	Small LDL (%)	0.378	<0.0001
Females	Large HDL (%)	−0.330	<0.0001
Model 3			
Males	VLDL (%)	0.425	0.006
Model 4			
T2D	Large HDL (%)	−0.430	0.006
Model 5			
NDO	Small LDL (%)	0.451	0.002
Model 6			
Controls	Small HDL (%)	0.448	0.007

Abbreviations: HDL, high-density lipoprotein; NDO, nondiabetic obese; LDL, low-density lipoprotein; T2D, patients with type 2 diabetes; VLDL, very low-density lipoprotein. All backward stepwise multiple regression models included the following variables: body mass index, hemoglobin A1C, triglyceride, VLDL (%), IDL (%), large LDL (%), small LDL (%), large HDL (%), and small HDL (%).

## Data Availability

All data generated or analyzed during this study are included in this published article. All data generated or analyzed during the current study are available from the corresponding author on reasonable request.

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
