# Peer review of "Gender-Dependent Associations between Serum Betatrophin Levels and Lipoprotein Subfractions in Diabetic and Nondiabetic Obese Patients"

_ijms, 2023, doi:10.3390/ijms242216504_

Round 1

Reviewer 1 Report

Comments and Suggestions for Authors

In the present study, Lőrincz and colleagues investigated the potential association between betatrophin (also known as angiopoietin-like protein 8, ANGPTL8) and lipoprotein subfractions. The authors found that serum betatrophin concentrations were increased in obesity and type 2 diabetes compared to controls in female patients, but not in males. Moreover, strong correlations between atherogenic lipoprotein subfractions  and betatrophin levels were also detected, which were also gender-depents. This is an interesting, well-designed and timely study shedding light into the relevance of gender in the regulatory role of betatrophin in lipoprotein metabolism. Nonetheless, some specific points require to be amended.

Specific comments:

1.       Introduction, page 1, lines 37-38: as regards the important role of ANGPTL8/betatrophin in lipid metabolism, it is worth mentioning that circulating concentrations of betatrophin are increased in metabolic diseases associated with lipid disorders, such as dyslipidemia (Gómez-Ambrosi J et al. J Clin Endocrinol Metab 2016, PMID: 27472196) or non-alcoholic fatty liver disease (NAFLD) (Perdomo C et al, Int J Mol Sci 2021, PMID: 34884755).

2.       Introduction, page 1, lines 41-43: I would suggest to remove this sentence, since the original paper stating that betatrophin controls pancreatic β cell proliferation was retracted (Yi P et al, Cell 2017, PMID: 28038792).

3.       Results, Figure 1: the authors split the cohort according to sex and found that the increase in betatrophin levels in the context in obesity and type 2 diabetes is mainly detected in females. It could be interesting to show the mean betatrophin values in males and females of the cohort in order to analyse whether there is a sexual dimorphism in the circulating concentrations of this hormone.

4.       Discussion, page 8, lines 207-217: ANGPTL8 constitutes also a hepatokine that acts in an autocrine/paracrine manner that decreases intracellular triglyceride content in hepatocytes through the downregulation of transcription factors and enzymes involved in lipogenesis (Perdomo C et al, Int J Mol Sci 2021, PMID: 34884755).

Author Response

Response to Reviewer#1

Dear Reviewer,

Thank you for your positive review and supportive comments. We would like to reply to your comments point by point. The changes of the revised manuscript are marked by track changes.

  1. Introduction, page 1, lines 37-38: as regards the important role of ANGPTL8/betatrophin in lipid metabolism, it is worth mentioning that circulating concentrations of betatrophin are increased in metabolic diseases associated with lipid disorders, such as dyslipidemia (Gómez-Ambrosi J et al. J Clin Endocrinol Metab 2016, PMID: 27472196) or non-alcoholic fatty liver disease (NAFLD) (Perdomo C et al, Int J Mol Sci 2021, PMID: 34884755).

Response: Thank you for your comment. According to your suggestion, we added the above-mentioned literature data and references to the introduction (ln43-45, ref3-4).

  1. Introduction, page 1, lines 41-43: I would suggest to remove this sentence, since the original paper stating that betatrophin controls pancreatic β cell proliferation was retracted (Yi P et al, Cell 2017, PMID: 28038792).

Response: Thank you for your comment. We removed the sentence and citation from the text (ln41-42).

  1. Results, Figure 1: the authors split the cohort according to sex and found that the increase in betatrophin levels in the context in obesity and type 2 diabetes is mainly detected in females. It could be interesting to show the mean betatrophin values in males and females of the cohort in order to analyse whether there is a sexual dimorphism in the circulating concentrations of this hormone.

 Response: Thank you for your suggestion. Analyzing all subjects, we did not find significant difference regarding serum betatrophin levels in females and in males [18.76 (13.30-29.20) vs. 20.76 (12.80-26.40) ng/mL; p=0.87]. We added these results to the Results section (ln144-146) and demonstrated them on Figure 1d.

  1. Discussion, page 8, lines 207-217: ANGPTL8 constitutes also a hepatokine that acts in an autocrine/paracrine manner that decreases intracellular triglyceride content in hepatocytes through the downregulation of transcription factors and enzymes involved in lipogenesis (Perdomo C et al, Int J Mol Sci 2021, PMID: 34884755).

Response: Thank you for your comment. We completed the Discussion with the action of betatrophin as a hepatokine and its important regulatory role in hepatocytes (ln233-235).

Again, we would like to thank you for your expertise and time to improve our manuscript.

Reviewer 2 Report

Comments and Suggestions for Authors

In this manuscript, Lőrincz et al describe the result of a cohort study focusing on the correlation between serum betatrophin levels and lipoprotein fractions in people with T2D, nondiabetic obesity, and normal control. Their data include serum betatrophin levels in a relatively large number of people, which is a strong point of this study. However, their main argument that betatrophin measurement may help cardiovascular risk assessment is not sufficiently supported by their data. I would raise the following points that potentially strengthen their arguments:

1. In fig. 2, betatrophin levels are generally positively correlated with lipoproteins that are elevated in T2D whereas it is generally negatively correlated with lipoproteins that are decreased in T2D (Table. 2). Therefore,  betatrophin might be just reflecting lipoprotein fractional changes in T2D, and it does not have an additional predictive value for cardiovascular risk assessment. To show the significance of betatrophin measurement in CV risk assessment in each group of patients, the authors should analyze the correlation between betatrophin levels and the factors listed in Fig. 2 in each T2D/ nondiabetic obese/ nonobese group.

2. Regarding Fig. 3, showing the correlation between conventional CV risk factors (e.g., hsCRP, insulin, and HOMA-R) and LDL/HDL sizes is important to show how strongly betatrophin is correlated with lipoprotein sizes compared with other risk factors.

3. In the discussion, lines 235-244, it is still unclear why betatrophin is correlated with HDL size. A mechanistic argument for this possible relationship is welcome.

4. Figure legend and/or panel labeling in Fig. 3 seem to be incorrect. Probably, the legend is not correct.

5. Lines 41-43 referring to ref3, the reliability of this study has been viewed with skepticism by several following studies. This sentence should be changed fairly reflecting several independent studies.

Author Response

Response to Reviewer#2

Dear Reviewer,

Thank you for your positive review and supportive comments. We would like to reply to your comments point by point. The changes of the revised manuscript are marked by track changes.

  1. In fig. 2, betatrophin levels are generally positively correlated with lipoproteins that are elevated in T2D whereas it is generally negatively correlated with lipoproteins that are decreased in T2D (Table. 2). Therefore, betatrophin might be just reflecting lipoprotein fractional changes in T2D, and it does not have an additional predictive value for cardiovascular risk assessment. To show the significance of betatrophin measurement in CV risk assessment in each group of patients, the authors should analyze the correlation between betatrophin levels and the factors listed in Fig. 2 in each T2D/ nondiabetic obese/ nonobese group.

Response: Thank you for your comment. We agree with the reviewer, analyzing correlations between betatrophin with laboratory parameters and lipoprotein subfractions in each group separately may highlight the significance of betatrophin measurement in cardiovascular risk assessment. Therefore, we summarized subgroup analyses in Supplementary Table S1 and mentioned in the text (ln199-205). Our data may demonstrate the significance of betatrophin measurement in cardiovascular risk assessment in obese patients with T2D.

  1. Regarding Fig. 3, showing the correlation between conventional CV risk factors (e.g., hsCRP, insulin, and HOMA-R) and LDL/HDL sizes is important to show how strongly betatrophin is correlated with lipoprotein sizes compared with other risk factors.

Response: Thank you for your comment. We analyzed the correlations of conventional cardiovascular risk factors (age, BMI, CRP, insulin and HOMA-IR) and lipoprotein subfractions. The results are summarized in Supplementary Table S2 (ln206-210).

  1. In the discussion, lines 235-244, it is still unclear why betatrophin is correlated with HDL size. A mechanistic argument for this possible relationship is welcome.

Response: Thank you for your suggestion. We completed the Discussion, and the arguments for the relationship between the HDL metabolism and hypertriglyceridemia associated with diabetes and obesity are added (ln263-273).

  1. Figure legend and/or panel labeling in Fig. 3 seem to be incorrect. Probably, the legend is not correct.

Response: Thank you for your comment. The legend of Figure 3 was corrected (ln190-196).

  1. Lines 41-43 referring to ref3, the reliability of this study has been viewed with skepticism by several following studies. This sentence should be changed fairly reflecting several independent studies.

Response: The above mentioned paper was retracted (ref3, Yi P et al, Cell 2017, PMID: 28038792). Reviewer#1 also suggested the deletion of the questionable sentence and reference; therefore, we decided to remove this sentence from the introduction (ln41-43).

Again, we would like to thank you for your expertise and time to improve our manuscript.

Round 2

Reviewer 2 Report

Comments and Suggestions for Authors

The authors sufficiently provided additional data to address my points.